# Economic barriers to diagnostic equity: A multi-country analysis of patient costs for rapid SARS-CoV-2 testing in sub-Saharan Africa

Obinna Ekwunife[1,2]*, Collin Mangenah[3], Lucky Ngwira[4], Elizabeth Corbett[4,5], Karin Hatzold[6], Elvis Isere[7], John Bimba[7], Euphemia Sibanda[3,8], Frances M. Cowan[3,8], Godpower Omoregie[9], Gabrielle Bonnet[5]

1 Division of Population Health, Department of Medicine, University at Buffalo, New York, United States of America, 2 Department of Clinical Pharmacy and Pharmacy Management, Nnamdi Azikiwe University, Awka, Nigeria, 3 Centre for Sexual Health, HIV/AIDS Research (CeSHHAR), Harare, Zimbabwe, 4 Malawi-Liverpool-Wellcome Trust Clinical Research Programme, Blantyre, Malawi, 5 Faculty of Infectious and Tropical Diseases, London School of Hygiene and Tropical Medicine, London, United Kingdom, 6 Population Services International, Washington, District of Columbia, United States of America, 7 Zankli Research Center, Bingham University, Karu, Nigeria, 8 Department of International Public Health, Liverpool School of Tropical Medicine, Liverpool, United Kingdom, 9 Society for Family Health, Abuja, Nigeria

* ekwunife@buffalo.edu

## Abstract

While the acute phase of the COVID-19 pandemic has passed, understanding the economic barriers to diagnostic access remains critical for future pandemic preparedness and universal health coverage. Implementing efficient testing modalities is crucial to achieving optimal value for both clients and healthcare providers. This study examines the cost and affordability of various SARS-CoV-2 antigen rapid-diagnostic-test modalities in Nigeria, Malawi, and Zimbabwe from a client perspective, providing a blueprint for future diagnostic strategies in Sub-Saharan Africa. Testing was offered for free through professional testing and self-testing in government or NGO-led primary healthcare centers across all countries, and in community pharmacies and drug stores in Nigeria. Data were collected from October 2022 to May 2023 through a survey of a random sample of adults visiting participating sites. The survey collected patient costs, including transportation, medical and non-medical expenses, and productivity loss. Affordability was assessed by the incidence of catastrophic health expenditure (defined as costs exceeding 10% of household income). The unit patient cost of testing in Nigeria, Malawi and Zimbabwe was $4.2, $2.7 and $2.7, respectively. In Nigeria, testing in community pharmacies and drug stores was cheaper than in primary healthcare centers. Self-testing cost less than professional testing in Nigeria ($1.3 versus $9.8), but more in Zimbabwe ($3.2 versus $2.3). In Malawi, Nigeria and Zimbabwe 40.6%, 28.6%, and 5.7% of clients, respectively, faced catastrophic health expenditures. SARS-CoV-2 antigen testing imposes a significant financial burden on clients.

**Data availability statement:** The data underlying this study contain potentially sensitive participant information, including individual-level data on health-seeking behavior, income, household expenditure, and, in the case of Zimbabwe, data collected at Female Sex Worker clinics. Public sharing of the de-identified dataset is restricted to protect participant confidentiality and to comply with the conditions of ethical approval granted by the following Research Ethics Committees and Institutional Review Boards: the Malawi College of Medicine Research Ethics Committee (COMREC); the Bingham University of Health Sciences Ethics Committee and the Federal Capital Territory Health Research Ethics Committee (FHREC), Nigeria; the Medical Research Council of Zimbabwe (MRCZ); the World Health Organization Ethics Review Committee (WHO ERC); and the London School of Hygiene and Tropical Medicine Ethics Committee (LSHTM Ethics). Researchers interested in accessing the minimal dataset necessary to replicate the findings of this study may submit a formal data access request to the corresponding author, Obinna Ekwunife, at ekwunife@buffalo.edu. Requests will be reviewed in consultation with the relevant ethics committees and institutional bodies listed above. Requestors should specify the intended research purpose, their institutional affiliation, and any relevant ethical approvals they hold. Contact details for the oversight bodies are as follows: COMREC (Malawi): comrec@medcol.mw; FHREC (Nigeria): researcheth@fcthhss.abj.gov.ng; MRCZ (Zimbabwe): mrcz@mrcz.org.zw; WHO ERC: ersec@who.int; LSHTM Ethics: ethics@lshtm.ac.uk. All data shared will remain anonymized in accordance with the approved protocols. All other relevant data are within the paper and its Supporting Information files.

**Funding:** This study was supported by UNITAID through Population Services International (PSI). The funders were not involved in the study design, data collection or analysis, the decision to publish, or manuscript preparation. The authors are solely responsible for the content, which does not necessarily reflect the official positions of UNITAID or PSI.

**Competing interests:** The authors have declared that no competing interests exist.

Even "free" testing carries high indirect costs that threaten diagnostic equity. Diversified testing modalities, such as community pharmacies and drug stores, may offer lower-cost options for sustainable diagnostic integration.

## Background

The coronavirus disease (COVID-19) resulting from severe acute respiratory syndrome coronavirus 2 (SARS-CoV-2) infection has led to significant mortality and morbidity since its initial reporting in December 2019. While the global health emergency has transitioned, the lessons learned regarding diagnostic infrastructure remain vital for health system strengthening. Testing, along with isolation, contact tracing, and the use of personal protective equipment (PPE), was essential to the response. While nucleic acid amplification testing (NAAT) is the diagnostic reference, antigen-detection rapid diagnostics tests (Ag-RDTs) are also recommended due to their ease of use and rapid results. [1] Ag-RDTs are particularly suited for resource-constrained settings since shortages of NAAT testing supplies, limited skilled laboratory personnel, high costs, and logistical challenges are common. The practice and knowledge related to SARS-CoV-2 Ag-RDTs could serve as a foundation for the timely identification of other endemic diseases and new variants. [2]

Guaranteeing affordable diagnostic access for everyone in sub-Saharan Africa is essential to prevent access disparities based on socioeconomic status. In the post-pandemic era, empirical data on patient-side costs is required to design sustainable health policies. In contexts where patients incur out-of-pocket expenses, these costs can discourage individuals from getting tested. Affordable testing diminishes these barriers, encouraging individuals to pursue testing without the weight of financial burden.

The existing literature on the patient-side economic burden of SARS-CoV-2 Ag-RDTs in sub-Saharan Africa is scarce. A study examined the cost and cost-effectiveness of implementing SD Biosensor antigen-detecting SARS-CoV-2 diagnostics tests in Kenya but estimated this cost for testing in a hospital only [3]. Another study looked at the health system costs of SARS-CoV-2 testing in Mozambique [4]. However, there is a lack of multi-country evidence comparing different delivery modalities, such as community pharmacies versus public clinics, from the patient's perspective.

In this study, we sought to understand which Ag-RDT testing modalities may be lowest-cost and thus most affordable for the majority of the population in three different sub-Saharan African settings that share a need for expanded community-based SARS-CoV-2 testing and persistent challenges in integrating such testing into routine public health services: Nigeria, Malawi, and Zimbabwe. Specifically, we aimed to assess (i) the cost incurred by patients per SARS-CoV-2 Ag-RDT test in different healthcare settings from the patient's perspective, and (ii) the affordability of the tests in these settings. By providing a retrospective analysis of these costs, this study helps understand the socioeconomic barriers to diagnostic uptake and the patient-side costs of different rapid testing modalities in Sub-Saharan Africa.

## Methods

### Study overview

This study, implemented by various health development partners and supported by Ministries of Health (MOH) in Nigeria, Malawi, and Zimbabwe, evaluated the implementation of SARS-CoV-2 Ag-RDT strategies including self-testing and linkage to treatment and prevention for symptomatic participants and asymptomatic contacts who were contacts of a known case. The implementation projects aimed to evaluate the professional use of SARS-CoV-2 Ag-RDTs for diagnostic testing of symptomatic patients presenting to primary care facilities (Nigeria, Malawi), Female Sex Worker Clinics (Zimbabwe), PHS New Start Centers (Zimbabwe) or community pharmacies/drug stores (Nigeria). Community pharmacies are licensed retail pharmaceutical outlets operated by trained pharmacists and authorized to dispense prescription and over-the-counter medicines. Drug stores (also referred to as patent and proprietary medicine vendors in some settings) are retail outlets that sell over-the-counter medicines and basic health commodities but are typically staffed by non-pharmacist personnel and operate under more limited regulatory scope. SARS-CoV-2 Ag-RDTs self-testing was also evaluated in all primary healthcare facilities in Zimbabwe and both community and drug stores in Nigeria (Table 1).

### Study design and sample

Participants were recruited between October 2022 to May 2023 among adults ≥ 18 years visiting participating health centers, pharmacies and drug stores. Sampling was undertaken as follows: 1) a sample of days was first defined ensuring representation of both weekdays and weekends, as those are expected to cater to statistically different populations; for sites where half-day visits were planned, we ensured balanced sampling of both mornings and evenings, 2) we approached either all adults in need of SARS-CoV-2 testing at the center or, if the flow of visitors to the institution was too important, one in two or one in three presenting patients. This ensured random sampling of the patients attending participating institutions. Sites were chosen in collaboration with local health authorities to represent common service delivery settings for SARS-CoV-2 testing in each country. In Zimbabwe, all static female sex worker clinics and all new start clinics undertaking testing were actually included, as their number was sufficiently limited.

### Cost analysis

We conducted a descriptive economic cost analysis of SARS-CoV-2 Ag RDT testing from the client's perspective, excluding health system costs. Our primary outcomes included the cost per SARS-CoV-2 test and the cost per positive SARS-CoV-2 case. The cost represented the 2023 value. We converted local currencies to US$ equivalents using each country's 2023 average exchange rate, sourced from the World Bank website [5]. No formal statistical comparisons were conducted across countries or testing modalities due to differences in study design, sampling approaches, and contextual heterogeneity.

Client costs of SARS-CoV-2 testing included transportation costs, medical costs, non-medical costs, and productivity loss due to testing (S1 File). Transportation costs consisted of the expenses incurred for traveling to the testing facility and back home or to the client's next destination. Medical costs included any expenses paid at the testing facility to access

**Table 1. SARS-CoV-2 Ag-RDT strategies in the different countries.**

| Use case | Country | Setting | Type of test |
|---|---|---|---|
| Health Centre | Malawi | Primary Healthcare | Professional use |
| | Nigeria | Primary healthcare | Professional use |
| | Zimbabwe | Primary healthcare | Professional use and self-test |
| Pharmacy or Drug store | Nigeria | Community pharmacy | Professional use and self-test |
| | | Drug store | Professional use and self-test |

testing, such as consultation fees, registration fees, and test kits. Non-medical costs included any additional expenses incurred by the client, aside from the medical costs, like buying extra food, supplies, or accommodation.

Productivity loss was estimated based on the total time spent accessing SARS-CoV-2 testing, including waiting time, time spent undergoing the test, time to receive results, and any additional time required to obtain results. These time components were summed and converted into hours. The monetary value of productivity loss was calculated using a combination of self-reported income loss and standardized estimates. For participants who reported taking time off work, we included self-reported income lost due to testing and any payments made to substitute labor. For Zimbabwe in particular, we replaced self-reported lost earnings due to SARS-CoV-2 Ag RDT testing with calculated lost earnings based on their monthly salary to address the substantial missing data. In addition, for accompanying persons, we applied a standardized hourly wage derived from the national minimum wage to estimate opportunity costs. The total productivity loss per participant was computed as the sum of lost income, payments for substitute labor, and estimated losses associated with accompanying individuals.

Furthermore, we assessed the cost of testing in different use cases (i.e., scenario analysis), such as primary healthcare centers, pharmacies, and drug stores. We presented the cost of SARS-CoV-2 testing as the cost per client and the cost per positive case. To calculate the cost per positive case, we divided the total cost by the number of positive cases identified through testing.

All observed cost values were retained in the analysis, including extreme values, to reflect real-world variability in patient-incurred costs. Consequently, some estimates, particularly in Nigeria, showed large standard deviations, reflecting substantial heterogeneity in time and income losses rather than data anomalies.

## Affordability assessment

We assessed the affordability of SARS-CoV-2 Ag-RDTs (for professional use and self-testing) in various use cases using catastrophic health expenditure approach. For each individual tested, we calculated the ratio of the cost per person tested with SARS-CoV-2 Ag-RDTs to the person's total household income. A cost exceeding 10% of household income would indicate catastrophic health expenditure. [6,7] All analyses were performed using Microsoft Excel and, additionally, using Stata software, version 17.0, StatCorp LLC, USA.

## Ethical considerations

This cross-country cost analysis was conducted as part of larger trials implemented with approval from the Ministries of Health in Malawi, Nigeria, and Zimbabwe. Ethical clearance for the parent studies was obtained from multiple review bodies, including the Malawi College of Medicine Research Ethics Committee (P.05/22/3649), the Bingham University of Health Sciences Ethics Committee and the Federal Capital Territory Health Research Ethics Committee in Nigeria (FHREC/2022/01/29/09-03-22), the Medical Research Council of Zimbabwe (MRCZ/A/2872), as well as external approvals from the World Health Organization Ethics Review Committee (CERC.0163; CERC.0160; CERC.0165) and the London School of Hygiene and Tropical Medicine (26886, 26931, 26874). All data were anonymized prior to analysis to ensure confidentiality.

## Results

### Study demographic characteristics

A total of 644 clients participated in the study, with 199 in Nigeria, 106 in Malawi, and 339 in Zimbabwe. The median age of the participants ranged from 27 to 32 years. In all three countries, more than half of the participants were females, and the majority had completed senior secondary school and/or had some higher education. The participants represented a diverse range of occupations, with approximately 30% of them being unemployed in Malawi and Zimbabwe. The

proportion of the responding population declaring they were in an income-earning occupation (excluding unemployed, students, homemakers and retired individuals) was 75% in Nigeria, 63% in Malawi and 62% and 66% in Zimbabwe's PHS New Start Centers and Female Sex Worker clinics respectively. These figures are similar to the national figures for employment in the working age population in 2022–2023 in these countries, which were estimated at 78% and 80% in Nigeria in 2022 and 2023, at 64% in Malawi, and at 59% in Zimbabwe [8]. Among the participants, Nigerian respondents had the lowest monthly income (M = US$84.0, SD = US$87.0), while Zimbabwean participants had the highest monthly income (M = US$207.5, SD = US$344.4). However, in Nigeria, only 40% of the participants expressed concerns about food insecurity in the last week, while approximately 59% and 73% were worried about food insecurity in Zimbabwe and Malawi, respectively (Table 2).

### Cost of accessing SARS-CoV-2 Ag-RDTs test (patients' perspective)

The mean cost of accessing a SARS-CoV-2 Ag-RDT test was lower in Malawi (M = US$2.7, SD = US$3.8; 95% CI: 2.0–3.4) and Zimbabwe (M = US$2.7, SD = US$3.6; 95% CI: 2.3–3.1) compared to Nigeria (M = US$4.2, SD = US$16.0; 95% CI: 2.0–6.4). The relatively large standard deviation observed in Nigeria reflects substantial variability in patient-incurred costs, driven primarily by differences in travel expenses and productivity losses across testing locations. The cost per positive case of SARS-CoV-2 Ag-RDT testing was US$416.1 in Nigeria and US$915.3 in Zimbabwe. We were not able to calculate this cost in Malawi since none of the participants tested positive (Table 3).

### Scenario analyses

In Nigeria, testing at primary healthcare facilities was associated with a much higher cost (US$9.6) compared to drug stores (US$1.8) and community pharmacies (US$0.6). Additionally, professional testing was associated with a higher client cost compared to self-testing (US$9.8 vs. US$1.3) in Nigeria. Conversely, in Zimbabwe, professional testing was associated with lower client costs compared to self-testing (US$2.3 vs. US$3.2) (Table 4).

### Affordability of SARS-CoV-2 Ag RDT Testing

The assessment of the potential catastrophic impact of SARS-CoV-2 Ag RDT testing revealed that approximately 28.6% of the 199 individuals in Nigeria, 40.6% of the 106 individuals in Malawi, and 5.7% of the 339 individuals in Zimbabwe may experience catastrophic health expenditures to access SARS-CoV-2 Ag RDT testing (Fig 1).

In Nigeria, a higher percentage of participants at primary healthcare facilities potentially experienced catastrophic health expenditures (approximately 32%), compared to those who tested in drug stores (approximately 29%) or those who tested in private pharmacies (approximately 25%). In Nigeria again, a higher percentage of participants who chose professional testing potentially experienced catastrophic health expenditures compared to participants who opted for self-testing. The same situation was also observed in Zimbabwe (Table 5)

## Discussion

In our study, we found that clients spent between US$2.7 – US$4.2 to test for SARS-CoV-2. While these data reflect a specific epidemiological window (late 2022 to early 2023), they provide a critical baseline for understanding the hidden economic barriers that persist in post-emergency diagnostic systems. Malawi and Zimbabwe had the lowest client cost while Nigeria had the highest. The high cost per positive case detected (from US$416.1 in Nigeria to US$915.3 in Zimbabwe) underscores the economic challenge of maintaining testing programs as prevalence drops, arguing for the integration of SARS-CoV-2 testing into broader, multiplexed respiratory surveillance systems. Our findings also show that even within the "post-emergency" phase, a significant proportion of individuals in Nigeria, Malawi, and Zimbabwe faced catastrophic health expenditures to access Ag RDTs. While we applied a standard threshold of 10% of household income to define catastrophic health expenditure, this threshold may not fully capture the extent of financial hardship across

**Table 2. Demographic characteristics of respondents (N = 644).**

| | Nigeria (n = 199) | Malawi (n = 106) | Zimbabwe (n = 339) |
|---|---|---|---|
| **Facility type, n (%)** | | | |
| Primary Health Care Center | 69/199 (34.7%) | 106/106 (100%) | 339/339 (100%) |
| Community Pharmacy | 53/199 (26.6%) | – | – |
| Drug store | 77/199 (38.7%) | – | – |
| **Demographics** | | | |
| Age, median [IQR] | 27 [24 – 33] | 28 [23 –35] | 32 [25 –39] |
| Female, n (%) | 106/199 (53.3%) | 67/106 (63.2%) | 288 (84.7%) |
| **Education, n (%)** | | | |
| No education or preschool/nursery school only | 1 (0.5%) | 4/106 (3.8%) | – |
| Primary school | 2 (1.0%) | 39/106 (36.8%)[1] | 44/338 (13.0%) |
| Junior secondary school | 3 (1.5%) | | 105/338 (31.1%) |
| Senior secondary school | 56 (28.1%) | 56/106 (52.8%) | 161/338 (47.6%) |
| Post-secondary/tertiary education | 137 (68.8%) | 7/106 (6.6%) | 28/338 (8.3%) |
| **eEmployment status, n (%)** | | | |
| Unemployed | – | 34/105 (32.4%) | 105/332 (31.6%) |
| Household farming, livestock or fishing | 10/199 (5.0%) | 2/105 (1.9%) | 44/332 (13.3%) |
| Running or helping in the non-farm business | 35/199 (17.6%) | 34/105 (32.4%) | 4/332 (1.2%) |
| Formal employment | 69/199 (34.7%) | 13/105 (12.4%) | 48/332 (14.5%) |
| Casual or part-time work | 36/199 (18.1%) | 16/105 (15.2%) | 42/332 (12.7%) |
| Student | 46/199 (23.1%) | 5/105 (4.8%) | 7/332 (2.1%) |
| Others | 3/199 (1.5%) | 6/105 (5.7%) | 82/332 (24.7%) |
| **Income (US$), mean (SD)** | | | |
| Individual Income per month, | 84.0 (87.0)[2] | 118.4 (343.2)[3] | 207.5 (344.4)[4] |
| Household Income per month, | 117.2 (129.6)[2] | 219.0 (1,039.6)[3] | 302.2 (632.9)[4] |
| **Socioeconomic class, n (%)** | | | |
| Poorest | 74/199 (37.5%) | 69/105 (65.7%) | 155/339 (45.7%) |
| Middle class | 92/199 (46.2%) | 34/105 (32.4%) | 176/339 (51.9%) |
| Richest | 33/199 (16.5%) | 2/105 (1.9%) | 8/339 (2.4%) |
| **Food insecurity (past week), n (%)** | | | |
| Worried about food | 80/199 (40.2%) | 78/105 (73.6%) | 200/339 (59.0%) |

Percentages are based on country-specific denominators.

[1] Standard 1–8.

[2] US$ = NGN 460.

[3] US$ = MK1,026.44.

[4] US$ = ZWL$1000 or ZAR18.

diverse socioeconomic contexts. In low-income settings, even smaller expenditures may impose a substantial burden on households, suggesting that our estimates may underestimate the true economic impact of testing. Overall, these findings indicate that "free" testing at the point of care is insufficient for achieving diagnostic equity if indirect costs like transport and productivity loss remain unaddressed. These barriers are not unique to SARS-CoV-2 and will likely hinder the uptake of diagnostics for future pathogens of pandemic potential.

In relation to similar studies, the two other studies on SARS-CoV-2 Ag-RDT testing costs in sub-Saharan Africa primarily focused on health systems costs [3,4]. However, the study in Kenya also estimated patient costs for the

**Table 3. Cost of accessing SARS-CoV-2 Ag-RDTs test (patients' perspective).**

| | Nigeria (US$) | | Malawi (US$) | | Zimbabwe (US$) | |
|---|---|---|---|---|---|---|
| Cost per client | n/N | M (SD) | n/N | M (SD) | n/N | M (SD) |
| Travel cost | 199/199 | 1.9 (15.4) | 106/106 | 0.1 (0.1) | 339/339 | 1.6 (2.6) |
| Medical cost | 2/199 | 0.7 (0.4) | 0/106 | – | 2/339 | 0 |
| Non-medical cost | 2/199 | 1.1 (1.5) | 0/106 | – | 339/339 | 0.6 (1.3) |
| Productivity loss | 199/199 | 2.3 (4.5) | 106/106 | 2.6 (3.8) | 339/339 | 0.5 (1.3)* |
| *Average Cost per client* | *199/199* | *4.2 (16.0)* | *106/106* | *2.7 (3.8)* | *339/339* | *2.7 (3.6)* |
| *Cost per positive case* | *2/199* | *416.1* | | *NA[1]* | *1/339* | *915.3* |

Notes: n represents the number of patients experiencing the cost and N is the total number of patients across all modalities in the country. M (SD) correspond to the average and standard deviation of the costs incurred by patients incurring those costs (i.e., among the n patients).

[1] No positive case reported.

*We replaced self-reported lost earnings due to SARS-CoV-2 Ag RDT testing with calculated lost earnings based on their monthly salary to address the substantial missing data in the former.

SARS-CoV-2 Ag-RDT test, with a median cost of US$2.92 [3]. This cost aligns with our estimated patient cost in health facilities in Malawi (US$2.7) and Zimbabwe (US$3.3) but is lower than the patient cost estimated in Nigeria (US$4.2). Building the literature on patient costs of accessing SARS-CoV-2 rapid testing should help with future work on costs and cost-effectiveness in sub-Saharan Africa [9] . The cost of approximately US$2.7 to US$4.2 for a SARS-CoV-2 test can be a significant amount for individuals in countries where, at a poverty headcount ratio of US$2.15 per day and based on 2017 purchasing power parity, approximately 31% of Nigerians, 40% of Zimbabweans, and 68% of Malawians live below the poverty line [10]. Therefore, integrating client costs in decision-making regarding the choice of diagnostic modalities and locations is crucial to ensure equitable access to testing and design "people-centred" diagnostic networks.

Perhaps the most important finding for future health policy is that utilizing testing facilities beyond the traditional hospitals or primary health care centers, such as community pharmacies and drug stores, was associated with lower costs. Further review of the client cost details in Nigeria showed that travel cost and productivity loss were higher in primary health centers compared to community pharmacy or drug store. Retail pharmacies and drug stores are more abundant than clinics and are often situated in high-traffic areas, ensuring easy accessibility for the general population [11]. Moreover, it is well-documented that wait times in primary healthcare facilities can be lengthy. [12,13] leading to increased productivity losses. Diagnostic delivery at more accessible locations, such as private pharmacies and drug stores, can substantially reduce travel time and wait times for the client, a finding that could be adapted to other endemic diseases or future outbreaks.

It is important to note that the choice of testing method (self-testing or professional testing) is not consistently associated with lower costs. For example, self-testing was associated with lower costs in Nigeria but was associated with higher costs in Zimbabwe. One might expect that clients choosing self-testing would spend more time on the testing process, potentially resulting in increased opportunity costs and higher client expenses if other costs, such as travel costs or wait times, are similar. In Nigeria, the time lost due to testing was lower with self-testing because the majority of clients who opted for self-testing participated in the study at private pharmacies (41%) and drug stores (59%), which are closer to clients than healthcare centers (less than 1% of the study participants who opted for self-testing did so through primary healthcare facilities). Conversely, in Zimbabwe, the higher cost of self-testing compared to professional testing is primarily driven by higher travel cost, higher lost earnings due to testing, and higher non-medical costs. Client costs associated with testing are therefore not systematically higher or lower for professional vs. self-testers. Instead, they seem associated with other drivers, including the specific location and accessibility of testing (self-testing was more accessible than professional testing in Nigeria but not in Zimbabwe), the profile of the patients being tested (self-testers were slightly wealthier

**Table 4. Cost of testing at different use cases or different scenarios.**

| | Cost per client (US$) | | | | | |
|---|---|---|---|---|---|---|
| **Nigeria** | | | | | | |
| | Primary health care | | Community pharmacy | | Drug store | |
| | n/N | M (SD) | n/N | M (SD) | n/N | M (SD) |
| Travel cost | 69/199 | 4.3 (26.1) | 53/199 | 0.5 (1.3) | 77/199 | 0.6 (0.8) |
| Medical cost | 2/199 | 0.7 (0.4) | 0/199 | – | 0/199 | – |
| Non-medical cost | 2/199 | 1.1 (1.5) | 0/199 | – | 0/199 | – |
| Productivity loss | 69/199 | 5.3 (5.6) | 53/199 | 0.1 (0.6) | 77/199 | 1.1 (3.3) |
| *Average cost per client* | *69/199* | *9.6 (26.0)* | *53/199* | *0.6 (1.4)* | *77/199* | *1.8 (3.8)* |
| | Professional testing | | Self-testing | | | |
| | n/N | M (SD) | n/N | M (SD) | | |
| Travel cost | 67/199 | 4.4 (26.5) | 131/199 | 0.6 (1.0) | | |
| Medical cost | 2/199 | 0.7 (0.4) | 0/199 | – | | |
| Non-medical cost | 2/199 | 1.1 (1.5) | 0/199 | – | | |
| Productivity loss | 67/199 | 5.4 (5.7) | 131/199 | 0.8 (2.7) | | |
| *Average cost per client* | *67/199* | *9.8 (26.4)* | *131/199* | *1.3 (3.1)* | | |
| **Malawi** | | | | | | |
| | Professional testing | | | | | |
| | n/N | M (SD) | | | | |
| Travel cost | 106/106 | 0.1 (0.1) | | | | |
| Medical cost | 0/106 | – | | | | |
| Non-medical cost | 0/106 | – | | | | |
| Productivity loss | 106/106 | 2.6 (3.8) | | | | |
| *Average cost per client* | *106/106* | *2.7 (3.8)* | | | | |
| **Zimbabwe** | | | | | | |
| | Professional testing | | Self-testing | | | |
| | n/N | M (SD) | n/N | M (SD) | | |
| Travel cost | 200/339 | 1.4 (2.4) | 138/339 | 1.8 (2.8) | | |
| Medical cost | 1/339 | 0 | 1/339 | 0 | | |
| Non-medical cost | 200/339 | 0.5 (1.3) | 138/339 | 0.7 (1.4) | | |
| Productivity loss* | 200/339 | 0.4 (0.7) | 138/339 | 0.7(1.9) | | |
| *Average cost per client* | *200/339* | *2.3 (3.4)* | *138/339* | *3.2 (4.0)* | | |

Notes: n represents the number of patients experiencing the cost and N is the total number of patients across all modalities in the country. M (SD) correspond to the average and standard deviation of the costs incurred by patients incurring those costs (i.e., among the n patients).

*We replaced self-reported lost earnings due to SARS-CoV-2 Ag RDT testing with calculated lost earnings based on their monthly salary to address the substantial missing data in the former.

on average in Zimbabwe than professional testers, i.e., daily income of US$9 versus US$8 respectively) and other drivers including wait times (56 minutes for self-testers in Zimbabwe vs. 27 minutes for provider-testing), time spent getting familiar with the test and travel cost.

While testing in pharmacies and drug stores was cheaper in Nigeria, establishing public-private partnerships (PPP) with these providers to enhance SARS-CoV-2 testing accessibility demands careful consideration, as numerous concerns are associated with this arrangement. Successfully implementing PPPs can be challenging when there are no formal contracts with private providers outlining their roles, a lack of oversight and consequences for inadequate governing body capabilities, and financial viability concerns [14,15]. It is essential to acknowledge that in the latter case, private service

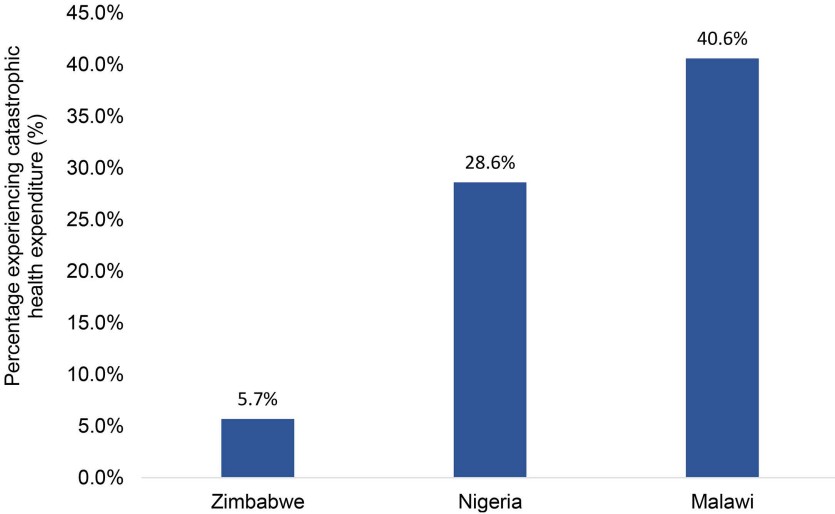

**Fig 1. The potentially catastrophic impact of SARS-CoV-2 Ag RDT testing.**

**Table 5. Percentage of persons that could potentially experience catastrophic health expenditure in the different socio-economic strata in Nigeria, Malawi, and Zimbabwe.**

|  | Nigeria | Malawi | Zimbabwe |
|---|---|---|---|
| *Socio-economic class* |  |  |  |
| Poorest (%) | 25/74 (33.8%) | 28/69 (40.6%) | 13/155 (8.4%) |
| Middle class (%) | 23/92 (25.0%) | 14/34 (41.2%) | 6/176 (3.4%) |
| Richest (%) | 9/33 (27.3%) | 1/2 (50.0%) | 0/8 (0.0%) |
| *Testing site* |  |  |  |
| Primary health care | 22/69 (31.9%) | – | – |
| Community pharmacy | 13/53 (24.5%) | – | – |
| Drug store | 22/77 (28.6%) | – | – |
| *Who tested* |  |  |  |
| Professional testing | 21/67 (31.3%) | 43/106 (40.6%) | 12/200 (6.0%) |
| Self-testing | 35/131 (26.7%) | – | 7/138 (5.1%) |

providers not employed by the government must be remunerated for the services they provide. These challenges were observed in several low- and middle-income countries (LMICs) during health sector reforms from the mid-1980s to the late 1990s in response to calls from the World Bank and the International Monetary Fund (IMF) to adopt PPPs for strengthening struggling health systems [15]. Notably, the PPP implemented in Tanzania to enhance healthcare delivery did not fully achieve the expected outcomes, contrary to popular belief [16]. Hence, each testing modality considered has to be carefully considered in a specific country context and properly overseen. More cost data collection in additional countries would help better understand the complexity of SARS-CoV-2 Ag-RDT client costs in different settings.

Although our analysis focused on cost and affordability, individuals seek SARS-CoV-2 testing for personal and public health benefits, including early care-seeking, informed self-isolation, reduced transmission, and reassurance before travel, social events, or work. Similarly, the decision not to test may derive from fear of restrictive measures and social harms, or misperceptions about COVID-19. Cost is only one dimension of access; improving uptake also requires raising awareness of benefits, increasing convenience, and aligning services with individuals' reasons for testing [17].

While this study provides important insights across three sub-Saharan African countries, future research involving a larger number of countries would be valuable to generate a more comprehensive regional picture of patient-level costs and affordability of diagnostic testing. Such broader analyses could further inform context-specific and region-wide strategies to improve equitable access to diagnostics.

This study presents a number of limitations. It utilized a limited number of test settings and was conducted in only three countries, which may limit the generalizability of findings across the broader sub-Saharan African region. While Nigeria, Malawi, and Zimbabwe were selected to reflect diverse implementation contexts, a larger multi-country study would provide a more comprehensive understanding of patient costs and affordability across different health systems in the region. We were not able to test the exact same testing modalities across all countries. For practical reasons, study sites were selected in collaboration with health authorities to be representative of SARS-CoV-2 testing setting; however, they did not constitute a random sample of all healthcare centres in the country. As a result, there may be some discrepancy between the patients in the targeted facility and all facilities. A more important potential bias relates to generalizability to the entire country's population. By design, we sampled individuals seeking care at health facilities and in need of SARS-CoV-2 testing. This excludes individuals who did not seek care, potentially due to affordability or access constraints. As a result, our findings likely underestimate the true economic burden of testing, particularly for more vulnerable populations who face greater barriers to accessing care. Consequently, our estimates of patient costs and affordability, including catastrophic health expenditure, should be interpreted as conservative. Individuals who are unable to access testing may incur even higher relative costs or forgo testing altogether due to financial barriers. For costs expressed in terms of working time lost, we may have underestimated loss of working time, and at the same time overestimated the income of those who may need SARS-CoV-2 testing (as people with lower incomes are less likely to seek care). In addition, productivity loss estimates relied partly on standardized assumptions based on minimum wage rates, particularly for accompanying individuals or where self-reported income data were unavailable. While this approach improves consistency, it may not fully capture individual-level variations in income. We did not conduct formal sensitivity analyses to assess the robustness of these assumptions. Furthermore, the high variability observed in cost data, particularly in Nigeria, reflects real-world heterogeneity but may limit comparability across settings. The survey was conducted between October 2022 to May 2023, and changes in public health trends since then may affect the applicability of the findings to current conditions. Finally, the use of a 10% threshold to define catastrophic health expenditure, while widely adopted, may not adequately reflect financial hardship across heterogeneous income groups, particularly among the poorest populations. In addition, household income data were self-reported and may be subject to recall bias or misreporting, which could affect the accuracy of catastrophic expenditure estimates.

Despite the limitations, the study provides valuable insights into the financial burden faced by individuals seeking SARS-CoV-2 testing and helps compare alternative testing scenarios that may offer cost-effective options for those in need of testing.

## Conclusions

This study provides retrospective yet vital knowledge for integrating diagnostics into routine primary care in LMICs. We demonstrate that the financial burden associated with SARS-CoV-2 Ag-RDT testing remains a threat to diagnostic equity. However, a promising solution lies in differentiated testing modalities. By expanding testing options to community pharmacies and drug stores, we can provide low-cost alternatives that are more accessible to a broader segment of the population. These settings offer convenience and affordability, as they are often conveniently located and equipped to provide testing services at competitive prices. This approach not only reduces the financial strain on individuals but ensures that as many people as possible have the opportunity to get tested, contributing to the collective effort to control the spread of the virus and protect public health. Any decision on testing modalities should however be informed by the specific country context and associated constraints.

## Supporting information

**S1 File. Patient cost questionnaire.**
(DOCX)

**S2 File. Inclusivity in global research questionnaire.**
(DOCX)

## Acknowledgments

The authors acknowledge the country study teams from Zankli Research Center in Nigeria, the Centre for Sexual Health, HIV and AIDS Research in Zimbabwe (CeSHAAR), and the Health Economics Policy Unit at the Kamuzu University of Health Sciences in Malawi that conducted the field study.

## Author contributions

**Conceptualization:** Elizabeth Corbett, Karin Hatzold, John Bimba, Godpower Omoregie, Gabrielle Bonnet.

**Data curation:** Collin Mangenah, Elvis Isere.

**Formal analysis:** Obinna Ekwunife, Collin Mangenah, Lucky Ngwira, Gabrielle Bonnet.

**Funding acquisition:** Lucky Ngwira, John Bimba.

**Investigation:** Euphemia Sibanda, Frances M. Cowan, Godpower Omoregie.

**Methodology:** Obinna Ekwunife.

**Project administration:** Lucky Ngwira, Karin Hatzold, Elvis Isere, John Bimba, Euphemia Sibanda, Frances M. Cowan, Godpower Omoregie.

**Resources:** Karin Hatzold.

**Supervision:** Elizabeth Corbett, Godpower Omoregie.

**Validation:** Elizabeth Corbett, Elvis Isere, Euphemia Sibanda, Frances M. Cowan, Gabrielle Bonnet.

**Writing – original draft:** Obinna Ekwunife.

**Writing – review & editing:** Obinna Ekwunife, Lucky Ngwira, Elizabeth Corbett, Karin Hatzold, Elvis Isere, John Bimba, Frances M. Cowan, Godpower Omoregie, Gabrielle Bonnet.

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
