## [Decision Letter · Decision Letter 0]

9 Apr 2026

PONE-D-26-08248Economic barriers to diagnostic equity: a multi-country analysis of patient costs for rapid SARS-CoV-2 testing in sub-Saharan AfricaPLOS One

Dear Dr. Ekwunife,

Thank you for submitting your manuscript to PLOS ONE. After careful consideration, we feel that it has merit but does not fully meet PLOS ONE’s publication criteria as it currently stands. Therefore, we invite you to submit a revised version of the manuscript that addresses the points raised during the review process.

**ACADEMIC EDITOR:** The study addresses a critical and understudied dimension of diagnostic equity by trying to quantify patient-level cost of SARS-CoV-2 testing across some African counties. The study is relevant for policy decisions and may be applicable to several African settings beyond the countries represented in the study. The methodology is well presented with some observations that require revisions to strengthen analytical rigor, clarify study design, and refine interpretation of findings. Please carefully review comments provided by Reviewer 2 and address them accordingly. Additionally, please add the following details:- Financial disclosure (funding details) – please add grant number- Ethical approval – add approvals number Please submit your revised manuscript by May 24 2026 11:59PM. If you will need more time than this to complete your revisions, please reply to this message or contact the journal office at plosone@plos.org. Please include the following items when submitting your revised manuscript:

We look forward to receiving your revised manuscript.

Kind regards,

Ibrahim Jahun, MD, MSC, PhD

Academic Editor

PLOS One

Journal Requirements:

“This study was supported by UNITAID through Population Services International (PSI). The funders were not involved in the study design, data collection or analysis, the decision to publish, or manuscript preparation. The authors are solely responsible for the content, which does not necessarily reflect the official positions of UNITAID or PSI.”

“This study was supported by UNITAID through Population Services International (PSI). The funders were not involved in the study design, data collection or analysis, the decision to publish, or manuscript preparation. The authors are solely responsible for the content, which does not necessarily reflect the official positions of UNITAID or PSI.”

5. We note that you have referenced “An unpublished study” which has currently not yet been accepted for publication. Please remove this from your References and amend this to state in the body of your manuscript: as detailed online in our guide for authors

6. We note that there is identifying data in the Supporting Information file “Dataset.xlsx”. Due to the inclusion of these potentially identifying data, we have removed this file from your file inventory. Prior to sharing human research participant data, authors should consult with an ethics committee to ensure data are shared in accordance with participant consent and all applicable local laws.

-Location data

Please remove or anonymize all personal information “Dates and Patient ID”, ensure that the data shared are in accordance with participant consent, and re-upload a fully anonymized data set. Please note that spreadsheet columns with personal information must be removed and not hidden as all hidden columns will appear in the published file.

Reviewers' comments:

Reviewer's Responses to Questions

**Comments to the Author**

1. Is the manuscript technically sound, and do the data support the conclusions?

Reviewer #1: Yes

Reviewer #2: Yes

2. Has the statistical analysis been performed appropriately and rigorously? 

Reviewer #1: Yes

Reviewer #2: Yes

3. Have the authors made all data underlying the findings in their manuscript fully available?

Reviewer #1: Yes

Reviewer #2: Yes

4. Is the manuscript presented in an intelligible fashion and written in standard English?

Reviewer #1: Yes

Reviewer #2: Yes

5. Review Comments to the Author

Reviewer #1: The manuscript was well written with data and results well presented. The authors explained well why the study is important in planning for future outbreaks and were able to show what was obtained in three African countries. There may be need for a much broader study involving more countries so as to have a good picture of what was obtained in sub-Saharan Africa and solutions proffered.

Reviewer #2: This manuscript addresses an important and underexplored aspect of diagnostic access in sub-Saharan Africa, namely the patient-side economic burden of SARS-CoV-2 rapid diagnostic testing. The multi-country design and focus on affordability provide valuable insights for health policy and future pandemic preparedness. Overall, the study is timely, relevant, and contributes meaningfully to the literature.

The manuscript is generally well-structured, and the conclusions are broadly supported by the data. However, several areas require clarification and minor strengthening to improve transparency, rigor, and interpretability.

Review comments

1. Clarity of costing methodology

While the study outlines the components of patient costs (transport, medical, non-medical, productivity loss), further clarification is needed on:

• How productivity loss was calculated in practice, particularly the substitution of missing self-reported income with minimum wage estimates

• Whether sensitivity analyses were conducted to assess the robustness of these assumptions

• How extreme values (e.g., very high SD in Nigeria costs) were handled

Given that productivity loss is a major cost driver, more transparency here is essential.

2. Statistical reporting and rigor

The analysis is largely descriptive, which is acceptable given the study objectives. However:

• No confidence intervals or uncertainty measures are presented for key estimates

• There is no indication of statistical comparison across countries or modalities

• The rationale for not performing comparative statistical tests should be stated

Including basic inferential comparisons or explicitly justifying their absence would strengthen the analytical rigor.

3. Interpretation of catastrophic health expenditure

The use of a 10% threshold is standard, but:

• The manuscript should acknowledge that this threshold may not fully capture financial hardship across heterogeneous income settings

• There should be a brief discussion of how income measurement (self-reported) may introduce bias

4. Generalizability

The limitations section correctly notes selection bias (only those who sought testing were included), but this point should be strengthened:

• The study likely underestimates true costs, especially for those unable to access care

• This limitation has important implications for interpreting affordability estimates

Other Comments

1. Typographical and formatting issues

• Minor inconsistencies (spacing, punctuation, duplicated percentage symbols in Table 2)

• Ensure uniform reporting of currency (US$ vs $)

2. Terminology consistency

• “Drug stores” vs “community pharmacies” should be clearly defined early and used consistently

3. Results clarity

• The very high standard deviation in Nigeria (SD = 16.0) should be briefly explained in the text

4. Figure and tables

• Figure 1 (supplementary files)could benefit from clearer labeling (x-axis titles and formatting)

• Some tables are dense and may need slight simplification for readability especially table 2

6. PLOS authors have the option to publish the peer review history of their article (what does this mean?). If published, this will include your full peer review and any attached files.

Reviewer #1: **Yes:** Vincent Pam Gyang

Reviewer #2: **Yes:** Ayoola Bosede

---

## [Author Response · Author response to Decision Letter 1]

20 Apr 2026

POINT BY POINT RESPONSE:

PONE-D-26-08248. Economic barriers to diagnostic equity: a multi-country analysis of patient costs for rapid SARS-CoV-2 testing in sub-Saharan Africa

ACADEMIC EDITOR:

1. please add the following details:

a - Financial disclosure (funding details) – please add grant number

Answer: We have added Unitaid grant number (2017-16-PSI-STAR to KH). See cover letter.

b - Ethical approval – add approvals number

Answer: We have added ethical approval numbers. See ethics considerations (pag3 8 – 9).

• A letter that responds to each point raised by the academic editor and reviewer(s). You should upload this letter as a separate file labeled 'Response to Reviewers'.

Answer: We uploaded a separate file labeled response to reviewers.

Answer: We uploaded a separate file labeled “Revised manuscript with track changes”.

Answer: We uploaded another unmarked version of the revised manuscript and labelled ‘Manuscript’

Answer: Not applicable.

Answer: Yes, the manuscript meets PLOS ONE’s style requirements.

Answer: Yes, we have filled the questionnaire and uploaded it as S2.

“This study was supported by UNITAID through Population Services International (PSI). The funders were not involved in the study design, data collection or analysis, the decision to publish, or manuscript preparation. The authors are solely responsible for the content, which does not necessarily reflect the official positions of UNITAID or PSI.”

“This study was supported by UNITAID through Population Services International (PSI). The funders were not involved in the study design, data collection or analysis, the decision to publish, or manuscript preparation. The authors are solely responsible for the content, which does not necessarily reflect the official positions of UNITAID or PSI.”

Answer: We have removed funding statement from the acknowledgement section. In our cover letter you will find the amended funding statement as follows: “This study was supported by UNITAID grant (2017-16-PSI-STAR to KH) through Population Services International (PSI). The funders were not involved in the study design, data collection or analysis, the decision to publish, or manuscript preparation. The authors are solely responsible for the content, which does not necessarily reflect the official positions of UNITAID or PSI.”

Answer: I have added my ORCID iD to my account.

5. We note that you have referenced “An unpublished study” which has currently not yet been accepted for publication. Please remove this from your References and amend this to state in the body of your manuscript: as detailed online in our guide for authors

Answer: The study cited was a manuscript on preprint server with citable DOI, which is acceptable by PLOSOne. Thus we have deled the word “unpublished”.

6. We note that there is identifying data in the Supporting Information file “Dataset.xlsx”. Due to the inclusion of these potentially identifying data, we have removed this file from your file inventory. Prior to sharing human research participant data, authors should consult with an ethics committee to ensure data are shared in accordance with participant consent and all applicable local laws.

-Location data

Please remove or anonymize all personal information “Dates and Patient ID”, ensure that the data shared are in accordance with participant consent, and re-upload a fully anonymized data set. Please note that spreadsheet columns with personal information must be removed and not hidden as all hidden columns will appear in the published file.

Answer: Thank you. We have ensured that we have anonymized all personal information in the manuscript and other documents uploaded.

Answer. We have included captions in the two supporting information files.

Answer: Noted

Answer: We have reviewed the reference list and ensured that it is complete and correct.

REVIEWER #1:

The manuscript was well written with data and results well presented. The authors explained well why the study is important in planning for future outbreaks and were able to show what was obtained in three African countries. There may be need for a much broader study involving more countries so as to have a good picture of what was obtained in sub-Saharan Africa and solutions proffered.

Response: Thank you for this comment. We appreciate the reviewer’s recognition of the importance of our multi-country analysis and its relevance for informing future outbreak preparedness. We have now strengthened the manuscript to explicitly acknowledge this limitation and to highlight the need for broader, regionally representative studies. We also emphasize that our findings provide a useful foundation for future large-scale, multi-country research aimed at informing diagnostic policy across sub-Saharan Africa. These revisions have been incorporated into the Discussion and Limitations sections of the manuscript as shown below:

Discussion:

“While this study provides important insights across three sub-Saharan African countries, future research involving a larger number of countries would be valuable to generate a more comprehensive regional picture of patient-level costs and affordability of diagnostic testing. Such broader analyses could further inform context-specific and region-wide strategies to improve equitable access to diagnostics.”

Limitation paragraph in Discussion:

“This study presents a number of limitations. It utilized a limited number of test settings and was conducted in only three countries, which may limit the generalizability of findings across the broader sub-Saharan African region. While Nigeria, Malawi, and Zimbabwe were selected to reflect diverse implementation contexts, a larger multi-country study would provide a more comprehensive understanding of patient costs and affordability across different health systems in the region.”

REVIEWER #2:

This manuscript addresses an important and underexplored aspect of diagnostic access in sub-Saharan Africa, namely the patient-side economic burden of SARS-CoV-2 rapid diagnostic testing. The multi-country design and focus on affordability provide valuable insights for health policy and future pandemic preparedness. Overall, the study is timely, relevant, and contributes meaningfully to the literature.

The manuscript is generally well-structured, and the conclusions are broadly supported by the data. However, several areas require clarification and minor strengthening to improve transparency, rigor, and interpretability.

Review comments

1. Clarity of costing methodology

While the study outlines the components of patient costs (transport, medical, non-medical, productivity loss), further clarification is needed on:

• How productivity loss was calculated in practice, particularly the substitution of missing self-reported income with minimum wage estimates

• Whether sensitivity analyses were conducted to assess the robustness of these assumptions

• How extreme values (e.g., very high SD in Nigeria costs) were handled

Given that productivity loss is a major cost driver, more transparency here is essential.

Response: Thank you for this insightful comment on the clarity of our costing methodology. We have revised the manuscript to provide a more detailed description of how productivity loss was calculated. Specifically, productivity loss was estimated based on the total time spent accessing testing (including waiting time, testing time, and time to receive results), which was converted into hours. This time was then monetized using either self-reported income loss (where available) and a standardized estimate based on the national minimum wage to account for accompanying persons (See page 6).

Regarding sensitivity analyses, given the descriptive nature of our study and reliance on observed field data, we did not conduct formal sensitivity analyses. This has now been explicitly stated as a limitation (see page 19).

Finally, we clarified that all observed values, including extreme values contributing to large standard deviations (e.g., in Nigeria), were retained to reflect real-world variability in patient costs rather than being truncated or excluded (see page 7).

These clarifications have been incorporated into the Methods and Limitations sections of the manuscript.

2. Statistical reporting and rigor

The analysis is largely descriptive, which is acceptable given the study objectives. However:

• No confidence intervals or uncertainty measures are presented for key estimates

• There is no indication of statistical comparison across countries or modalities

• The rationale for not performing comparative statistical tests should be stated

Including basic inferential comparisons or explicitly justifying their absence would strengthen the analytical rigor.

Response: Thank you for this valuable comment. We agree that presenting measures of uncertainty strengthens the interpretation of descriptive findings. In response, we have now included 95% confidence intervals for key cost estimates (cost per client) across countries, calculated using the reported means, standard deviations, and sample sizes (See page 11).

Regarding statistical comparisons across countries and testing modalities, our study was designed as a descriptive, multi-country implementation analysis rather than a hypothesis-testing study. Differences in sampling strategies, contextual factors, and heterogeneity in testing modalities across countries limit the validity of direct statistical comparisons. We have now clarified this rationale in the Method section (page 6).

3. Interpretation of catastrophic health expenditure

The use of a 10% threshold is standard, but:

• The manuscript should acknowledge that this threshold may not fully capture financial hardship across heterogeneous income settings

• There should be a brief discussion of how income measurement (self-reported) may introduce bias

Response: Thank you for this important comment regarding the interpretation of catastrophic health expenditure.

We agree that while the 10% threshold is widely used in the literature, it may not fully capture the extent of financial hardship across heterogeneous income settings. We have revised the manuscript to acknowledge this limitatio

---

## [Decision Letter · Decision Letter 1]

13 May 2026

Economic barriers to diagnostic equity: a multi-country analysis of patient costs for rapid SARS-CoV-2 testing in sub-Saharan Africa

PONE-D-26-08248R1

Dear Dr. Ekwunife,

We’re pleased to inform you that your manuscript has been judged scientifically suitable for publication and will be formally accepted for publication once it meets all outstanding technical requirements.

Kind regards,

Ibrahim Jahun, MD, MSC, PhD

Academic Editor

PLOS One

Reviewers' comments:

Reviewer's Responses to Questions

**Comments to the Author**

1. If the authors have adequately addressed your comments raised in a previous round of review and you feel that this manuscript is now acceptable for publication, you may indicate that here to bypass the “Comments to the Author” section, enter your conflict of interest statement in the “Confidential to Editor” section, and submit your "Accept" recommendation.

Reviewer #2: All comments have been addressed

2. Is the manuscript technically sound, and do the data support the conclusions?

Reviewer #2: Yes

3. Has the statistical analysis been performed appropriately and rigorously? 

Reviewer #2: Yes

4. Have the authors made all data underlying the findings in their manuscript fully available?

Reviewer #2: Yes

5. Is the manuscript presented in an intelligible fashion and written in standard English?

Reviewer #2: Yes

6. Review Comments to the Author

Reviewer #2: Thank you to the authors for the time and effort invested in revising the manuscript and responding carefully to all the concerns raised during the review process. The responses were clear, satisfactory, and adequately addressed the major and minor issues highlighted. The revisions have improved the clarity, methodological transparency, and overall quality of the manuscript. In my opinion, the paper is now substantially strengthened and is suitable for publication in PLOS ONE.

7. PLOS authors have the option to publish the peer review history of their article (what does this mean?). If published, this will include your full peer review and any attached files.

Reviewer #2: **Yes:** Dr. Ayoola O. Bosede

---

## [Editor Report · Acceptance letter]

PONE-D-26-08248R1

PLOS One

Dear Dr. Ekwunife,

I'm pleased to inform you that your manuscript has been deemed suitable for publication in PLOS One. Congratulations! Your manuscript is now being handed over to our production team.

Kind regards,

on behalf of

Dr. Ibrahim Jahun

Academic Editor

PLOS One